# Acute Administration of Caffeine: The Effect on Motor Coordination, Higher Brain Cognitive Functions, and the Social Behavior of BLC57 Mice

**DOI:** 10.3390/bs8080065

**Published:** 2018-07-25

**Authors:** Sayed Almosawi, Hasan Baksh, Abdulrahman Qareeballa, Faisal Falamarzi, Bano Alsaleh, Mallak Alrabaani, Ali Alkalbani, Sadiq Mahdi, Amer Kamal

**Affiliations:** Physiology Department, College of Medicine and Medical Sciences, Arabian Gulf University, P.O. Box 26671, Manama 1111, Bahrain; sayed_hashim8@hotmail.com (S.A.); Hasan.bakhsh@live.com (H.B.); Abdulrahman.altayeb@hotmail.com (A.Q.); faisal_ahmed95@hotmail.com (F.F.); Balsaleh94@gmail.com (B.A.); dr.malak95@outlook.sa (M.A.); Ali.elkalbani.95@gmail.com (A.A.); Sadiq.aldairi@gmail.com (S.M.)

**Keywords:** caffeine, cognition, motor coordination, memory, social behavior, mice

## Abstract

Excessive caffeine consumption causes adverse health effects. The effects of moderate and high doses of caffeine consumption on the motor coordination, cognitive brain functions, and the social behavior in mice were studied. Animals were divided into three groups: control group, moderate dose group (Ac MD), and high dose group (Ac HD). The animals were tested after 7 days of caffeine administration. A rotarod test for motor coordination showed that the mice of the moderate dose group could stay on the rotating rod longer before falling in comparison to the control group and the high dose group. A water maze test for learning and memory showed better performance of mice receiving the moderate dose of caffeine compared to the other groups. Animals that were administered moderate as well as high doses of caffeine showed no sociability and no preference for social novelty in the three-chamber test used to test social behavior. In an elevated plus maze test, control animals showed no anxiety-like behavior while mice from both of the groups administered with caffeine showed anxiety-like behaviors. Our data conclude that the effects of caffeine on higher brain functions depend on the administration dose. When caffeine was given in moderate doses, it resulted in enhancement of memory and motor coordination functions. However, high doses caused defects in memory and learning. The social behavior of the mice, as determined by the level of anxiety and sociability, was affected negatively by moderate as well as high dose caffeine administration.

## 1. Introduction

Caffeine, a widely consumed substance by man, is found in tea, coffee, energy drinks, and other beverages. Recently, it was estimated that 90% of the U.S. population and 80% of the world’s population consume caffeine on a daily basis [1]. It has been reported that caffeine can enhance memory in both animal models and humans [2,3]. While the human tolerance to moderate intakes of caffeine is good, heavy caffeine consumption could cause severe health effects [4].

Memory is significantly affected by sleep, in which the brain’s neural connections are strengthened. The strengthening of the neural connections enhances the brain’s ability to retain memory. During sleep, different parts of the brain process the memories and convert them into long-term memory. Insomnia or sleep deprivation results in a lower retention rate of memories due to the neural connections not being as strong as they need to be [5]. Sleep deprivation, in general, is associated with the deterioration of memory [6] as well as other negative outcome such as suicidal behavior [7].

Acute caffeine administration enhances learning and memory functions in rodents [8]. In addition, chronic caffeine administration has been shown to be beneficial in animals with Alzheimer’s diseases as well as in age-related defects in cognition [9]. However, caffeine can cause delayed sleep onset in both humans and rodents [10] as well as stimulation of locomotor activity. The effects of caffeine are believed to be mediated through the antagonism of adenosine receptors, especially A1R and A2AR, and it exerts a stimulating effect on locomotor activity at low to moderate doses. At higher doses, however, it has even depressive effects [11,12]. A significant correlation was found between the caffeine dose and the level of depressive illness [13]. Data from other research studies show a direct correlation between the adenosine homeostasis and mood disorders. Patients with major depression were found to have lower levels of serum adenosine deaminase compared to the control group, with a negative correlation between the enzyme activity and the severity of depression [14].

Due to the effect of caffeine on adenosine receptors and on multiple neurotransmitters that lead to arousal, high doses of caffeine can cause a dysfunctional arousal state that causes sleep disturbances that include alteration of sleep quality, latency, and total sleep time [15].

Although the specific mechanism is not yet fully understood, multiple research studies have successfully linked sleep deprivation with anxiety disorder [16,17].

Anxiety disorder, which is a mental disorder characterized by out of proportion feelings of fear and anxiety in frequency and/or duration relative to the actual situation [18,19,20]. Interestingly, the effects of caffeine on anxiety differ based on the dose ingested. It has been observed that high doses can cause anxiety, whereas low doses can have anxiolytic effects [21,22,23,24].

This study aimed to measure the effect of caffeine on learning, memory, anxiety, and social behavior in mice. We hypothesize that administering moderate doses of caffeine increases the higher brain functions and locomotion of mice. Memory enhancement could be due to an inhibition of adenosine A_1_ receptors that strongly inhibit the release of acetylcholine from pyramidal hippocampal neurons, while locomotor enhancement could be attributed to the multiple effects of caffeine on skeletal muscle contraction (by either modulating the calcium homeostasis in the muscle fibers and/or increasing the sensitivity of myofilaments to calcium ions). Furthermore, caffeine-treated mice will display anxiogenic behavior and a decrease in social parameters, which can be caused by a blockade of benzodiazepine binding sites on GABA_A_ receptors, stimulation of central noradrenergic activity, or antagonism of adenosine receptors.

## 2. Materials and Methods

### 2.1. Animals

Adolescent male BLC57 mice were bred in the animal department of the Arabian Gulf University. They were housed in cages on sawdust. The animals were divided into the following: control group (*n* = 8, were given normal water), acute moderate dose (*n* = 8, were given moderate dose caffeine (0.1 g/L, i.e., 20 mg/kg) for 1 week, Ac MD), and acute high dose (*n* = 8, were given high dose caffeine for 1 week, Ac HD 1 g/L, i.e., 200 mg/kg). These doses are comparable to other previous studies [25]. Each of the groups was tested in rotarod and water-maze in one day with three-hour intervals. The elevated plus maze and three-chamber social apparatus tests were done the next day with same intervals. All tests were performed when the animals were six weeks of age. All experiment procedures followed the animal care ethics of Arabian Gulf University, Manama, Bahrain.

### 2.2. Tests

#### 2.2.1. Rotarod (RR)

The accelerating rotarod assesses motor coordination and balance. Mice were placed on a cylinder that rotates at a pre-assigned speed of 45 revolutions per minute (rpm). Each mouse was habituated for a total of a 6-min period divided into three trials of 2 min. During each trial, the mouse was replaced on the rod when it fell. Habituation was not recorded. Following the habituation, the test was performed with three trials per test. Latency to fall from the rotating rod was recorded. The three groups of mice were tested within the same experiment to allow comparison of baseline motor performance. Habituation and testing were done on the same day for each group.

#### 2.2.2. Elevated Plus Maze Test (EPM)

The elevated plus maze tests anxiety-like behavior [24]; it consists of two open arms (25 cm × 5 cm) and two enclosed arms of the same size at opposite sides of each other. The enclosed arms are surrounded by 15 cm high walls. The edges, 3 mm high, surround the open arms, minimizing the likelihood of animals falling from the apparatus. Both arms are 55 cm above the floor. Between the arms is a central square area (5 cm × 5 cm) where the mouse is placed.

The entire apparatus was cleaned using 70% ethanol between each subject. Each mouse was individually placed in the central square of the maze and was allowed to freely explore the apparatus. The mice’s behavior was recorded for the test period of 10 min and then analyzed. The number of entries per arm and the time spent in the open arms were recorded. An entry is recorded when all four paws enter the arm. The numbers of entries and time spent in the open arms reflect the general behavior of the mice. The less common entrance of the mice into the open arms of the maze, as well as the decreased amount of time spent in them, was considered as anxiety-like behavior.

#### 2.2.3. Morris Water Maze Test (MWM)

A water maze measures spatial learning and memory [26,27]. The apparatus consisted of a circular swimming pool (140 cm diameter and 50 cm height, filled to a depth of 30 cm); the water was maintained at room temperature (26 °C–28 °C). The maze was housed in a darkened room with visual cues and illuminated by sparse red light. It was divided into four equal quadrants by two diagonal lines set by the program.

Each mouse was given five acquisition trials per day for the first day (training day) to learn the position of a hidden ‘escape’ platform, which is submerged 2 cm below the water surface, at a fixed location inside the pool. On each trial, the mice were released from one of four predetermined positions on the perimeter of the pool. Animals were given a maximum of 120 s to find the platform and were allowed to remain on the platform for 20 s. Mice that failed to locate the platform were put onto it by the experimenter and allowed to stay there for 20 s.

The position and movement of the animals, in the pool, was captured and analyzed every 0.2 s, using a video-camera computer system, and ANY-maze video- tracking system (Stoelting Co., Wood Dale, IL, USA). Outcome measures were latency time and distance swum to reach the platform. Performance in each trial was averaged to yield one data point per mouse per test. The speed of swimming (a measure of motor function [26]) was measured as a control between the groups. Following the test, a probe test was performed in which the platform was removed, and each animal was allowed to swim for 120 s.

#### 2.2.4. Three-Chambers Social Apparatus (Crawley’s Sociability and Preference for Social Novelty Test) (3C)

The three-chambers social apparatus assesses sociability and preference for social novelty [28,29,30]. The rectangular three-chambered box consists of chambers with the dimensions of 20 cm × 40 cm × 22 cm. The walls of the box are made of clear Plexiglas. The dividing walls (also made of Plexiglas) had small rectangular openings (5 cm × 3 cm) allowing access to each chamber. Each chamber contained a cage which was 11 cm high, with a bottom diameter of 9 cm. The test consists of habituation for 5 min and two 10-min sessions. The subject mouse was first placed in the middle chamber to habituate for 5 min. Session 1 was started by placing an unfamiliar mouse (stranger 1) that had no prior contact with the subject mouse inside the wire cage in one of the side chambers. The subject mouse was then allowed to explore the entire apparatus freely. The time spent in each chamber, as well as the number of chamber entries, was recorded.

Session 2 was started by placing a second unfamiliar mouse in the wire cage inside the chamber that was empty during session 1. The test mouse was then left to freely choose between the chamber containing the already investigated mouse (stranger 1) and the one containing the novel unfamiliar mouse (stranger 2). The said strangers were of the same species and gender. The same parameters recorded for session 1 were recorded for session 2. The apparatus was cleaned with 70% ethanol between subjects. Session 1 tests for sociability, which is evident by the subject spending more time in the chamber containing a mouse than in the empty chamber. Preference for social novelty, measured in session 2, is indicated by spending more time in the chamber containing the novel mouse than the one containing the already investigated mouse.

## 3. Statistical Analysis

Data are presented as average ± SEM (standard error of mean) unless indicated otherwise. Comparisons between and within groups were made using ANOVA and post-hoc paired or unpaired two-tail *t*-tests. All statistical tests were performed with Microsoft Excel™ 2010 incorporating the Analysis Tool Pak add-in. Data were expressed as mean ± SEM. Statistical significance was set at a *p* value of less than 0.05.

## 4. Results

### 4.1. Rotarod Test: Ac MD Displayed Better Motor Coordination than the Other Groups

In comparison to the control (Cont; 29.83 ± 2.5 s) and Ac HD (26.2 ± 2.5 s) groups, Ac MD spent significantly more time on the rotating rod before falling (41.1 ± 4.3 s, ANOVA test, *p* < 0.05) (Figure 1).

### 4.2. Improved Performance Displayed by Ac MD Group in Morris Water Maze Test

Cognitive function was assessed by using a Morris water maze test (Figure 2). Latency (Figure 2A) to reach the platform of Ac MD group was significantly better (30.45 ± 7.3 s) compared to the control group latency (54.4 ± 6.8 s, ANOVA test, *p* < 0.05). However, the Ac HD group took significantly more time to reach the platform compared to the control group (73.4 ± 6.4 s, ANOVA test, *p* < 0.05). There were significant differences in the distances (Figure 2B) traveled by each group to reach the platform in which the Ac MD group traveled less distance (5.8 ± 0.96 dm) compared to the other groups (ANOVA test, *p* < 0.05). Another outcome that was noted was in the swimming velocity (Figure 2C), where the velocity of the Ac MD group was the highest (0.30 ± 0.02 m/s). The probe test (Figure 2D) revealed that the Ac MD group exhibited learning behavior, as they spent more time (36 ± 2.1%) in the disc zone than the other groups (ANOVA test, *p* < 0.05). The mice that spent minimal time in the disc zone were of the Ac HD group (30.2 ± 1.8%).

### 4.3. Increased Anxiety in Caffeine-Treated Mice

Anxiety was tested using the elevated plus maze test. Control animals showed less anxiety-like behavior by spending more time in the open arms compared to the other groups during the test. On the contrary, the caffeine-treated mice spent significantly less time in the open arms, which indicates an increase in anxiety (ANOVA test, *p* < 0.05, Figure 3). However, there was no significant difference between the Ac MD and Ac HD groups.

### 4.4. Lack of Sociability and Preference for Social Novelty in Caffeine-Treated Mice

In the three-chamber test, the control group animals showed normal sociability (session 1) by preferring to be in the chamber containing another mouse rather than staying in the empty chamber (315.4 ± 17.9 s versus 217 ± 22.4 s, respectively). They also showed a normal preference for social novelty (session 2) by spending more time in the chamber that contained a novel mouse than the chamber containing the old mouse (347 ± 38.1 s versus 184.8 ± 29.5 s, respectively). On the other hand, caffeine-treated animals demonstrated a lack in both sociability (session 1) and preference for social novelty (session 2), with no significant difference between the two treated groups (Figure 4).

## 5. Discussion and Conclusions

Caffeine affects many body functions including those of the cardiovascular, respiratory, renal, and nervous systems. Among the many mechanisms by which it acts, caffeine is known to antagonize adenosine and benzodiazepine receptors and essential enzymes like phosphodiesterase. In addition, it has been shown to inhibit the release of calcium ions from intracellular stores.

In relation to higher brain functions and social behavior, the antagonism of adenosine receptors is the most important mechanism. Competitive binding of caffeine to adenosine receptors resulted in the modulation of most of the central nervous system (CNS) neurotransmitters release including norepinephrine, dopamine, glutamate, acetylcholine, gamma-aminobutyric acid (GABA), and numerous others [31].

Adenosine receptors are mainly of two types: A_1_ and A_2_; both are nonselectively antagonized by caffeine. However, the specificity of inhibition depends on the concentration in the blood [32].

The research in hand investigates the effects of different doses of caffeine on four behavioral parameters in mice. There was previously a lack of research studying this number of variables under the effect of different doses of caffeine for short and long durations. Thus, the idea for this research came to life. The different parameters measured—motor coordination, spatial memory, anxiety, and social behavior—showed variations in results. The performance of the Ac MD group gave the most robust results concerning spatial memory and motor function, whereas the Ac HD group gave the weakest. Furthermore, caffeine increased anxious behavior and decreased sociability and the preference for social novelty.

According to the results collected and analyzed, mice treated with moderate doses of caffeine showed enhanced motor and spatial memory functions, although they had the most anxious behavior, decreased sociability, and decreased preference for social novelty. On the other hand, mice treated with high doses of caffeine showed deterioration in motor and spatial memory functions, with increased anxious behavior and similarly decreased sociability and social novelty as the group mentioned above.

Based on other research, the effects of caffeine on motor function are highly dose dependent. In fact, caffeine has biphasic effects, in which low doses increase motor function while high doses decrease it [27,28,29,30,33]. High caffeine concentration affects skeletal muscle contraction by either modulating the calcium homeostasis in the muscle fibers [34] and/or increasing the sensitivity of myofilaments to calcium ions [35].

Low doses of caffeine selectively decrease the activity in the striatum and nucleus accumbens [36,37,38]. This suggests that low doses of caffeine, similar to that of typical human caffeine consumption, are stimulating due to the inhibitory effect on adenosine receptors that are abundant in striatum.

The research at hand showed similar results; mice treated with moderate doses of caffeine showed enhancement in motor functions, while those treated acutely with high doses showed the opposite.

When examining the effects on spatial memory, it was apparent that low doses of caffeine enhance performance while high doses deteriorate it. In fact, studies have concluded that activation of adenosine A_1_ receptors strongly inhibits the release of acetylcholine from pyramidal hippocampal neurons [39,40,41]. Acetylcholine has been known to be important for memory storage [42]. Adenosine receptor knockout studies have demonstrated defects in cognition, memory, and social behavior [43]. In addition, synaptic plasticity of the hippocampus in the form of long-term potentiation (LTP) and depression (LTD) and higher brain functions were shown to be modulated by adenosine receptors and consequently by adenosine receptors antagonism [44].

To our knowledge, the literature contains few pieces of research on the effect of caffeine on memory retrieval, a topic that deserves further questioning. The results obtained in previously existing studies agree with our results that caffeine improves retention of cognitive function on short-term ingestion when given in lower doses [45].

Our results support previous studies in that lower doses of caffeine enhance spatial memory function. In fact, the AC MD group had significant improvement in memory retrieval, while the AC HD group was negatively affected compared to the control group [46].

Caffeine administration can exert anxiolytic or anxiogenic effects in rodents depending on the anxiety test employed, the rat strain, and the sex of the rat [47]. Recent reports have associated caffeine with anxiety-related behaviors, and the suggested mechanisms include: blockade of benzodiazepine binding sites on GABA_A_ receptors, stimulation of central noradrenergic activity, or antagonism of adenosine receptors [24,26,27]. Since adenosine is involved in the inhibition of cholinergic neurons in the brain stem, caffeine is thus capable of affecting acetylcholine actions and consequently arousal and electroencephalogram (EEG) wave patterns [48]. Arousal is also maintained and regulated by other neurotransmitters, especially dopamine and norepinephrine. Since caffeine can affect the availability of these neurotransmitters by its action on A1 adenosine receptors, it is postulated that this could be another mechanism to affect arousal. In fact, high doses of caffeine can affect brain function through the modulation of many other neurotransmitters by the same mechanism.

Our research showed anxiogenic effects of caffeine on the treated mice; this finding is in opposition to the idea that high doses of caffeine exert an anxiogenic effect while low doses exert an anxiolytic effect [49]. In fact, there was no clear relationship between the dose and the effect, as the Ac MD group was more anxious than the Ac HD group.

There is a lack of research concerning caffeine and its effects on sociability and preference for social novelty. Several studies showed that caffeine decreased social interaction in mice and rats [50,51]. The suggested mechanism can be related to its anxiogenic actions [50,51,52]. Another study demonstrated that a high dose of caffeine decreases the level of social interaction [53].

In the three-chamber test, the control group that was treated with saline solution spent more time exploring the conspecific than the unknown object. This pattern of behavior was preserved after administration of the moderate dose of caffeine but not after the high dose, which further indicates a lack of preference for the conspecific after receiving the high doses of caffeine. The same study showed that the high doses of caffeine significantly decreased the time spent sniffing the novel conspecific. As a result, had high dose of caffeine reduces the sociability and preference for social novelty [54].

Our study demonstrates that the acutely treated groups of mice were less social than the control group. Such results indicate that the decrease in sociability is not significantly dose-dependent. Similar results were shown regarding preference for social novelty in previous pieces of research. However, the Ac HD group preferred to spend more time with the novel mice than the Ac MD group. However, it should be mentioned that our results were based upon considering caffeine use on adolescent male mice. Different outcome results cannot be excluded by studying the effects of caffeine on female mice or younger or older age groups.

Overall, this study documents the diverse effects of caffeine on different parameters. The Ac MD group showed better results in motor and spatial memory functions, whereas the Ac HD group was less anxious than the Ac MD group. The most striking result was that sociability was not dose-dependent but had an equal effect on mice when administered. Since there is no concrete evidence that proves the link between the dose of caffeine and the level of decrease in sociability, further investigation is advised.

## Figures and Tables

**Figure 1 behavsci-08-00065-f001:**
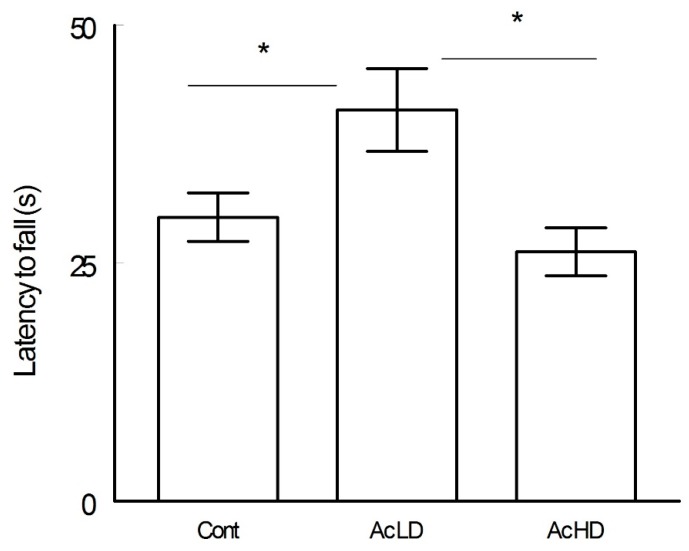
Latency to fall (mean ± SEM seconds) in the rotarod test for the control, moderate dose, and high dose groups. The time spent by Ac MD was significantly more than the other groups (ANOVA test, *p* < 0.05). Cont: control; Ac MD: moderate dose group; Ac HD: high dose group. Asterisks mark significant differences between the groups.

**Figure 2 behavsci-08-00065-f002:**
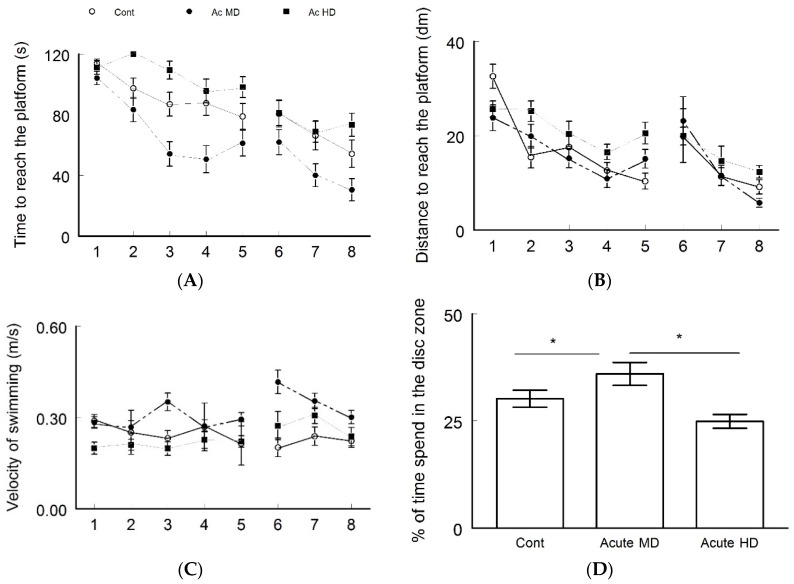
Morris water maze test used to assess cognitive function of the control, moderate dose, and high dose groups. (**A**) Time (latency) to reach the platform; (**B**) Distance to reach platform; (**C**) Velocity of swimming (m/s); and (**D**) % of time spent in the disc zone (Probe Test). In all the parameters tested, the Ac MD group showed better results than the other groups (ANOVA Test, *p* < 0.05).

**Figure 3 behavsci-08-00065-f003:**
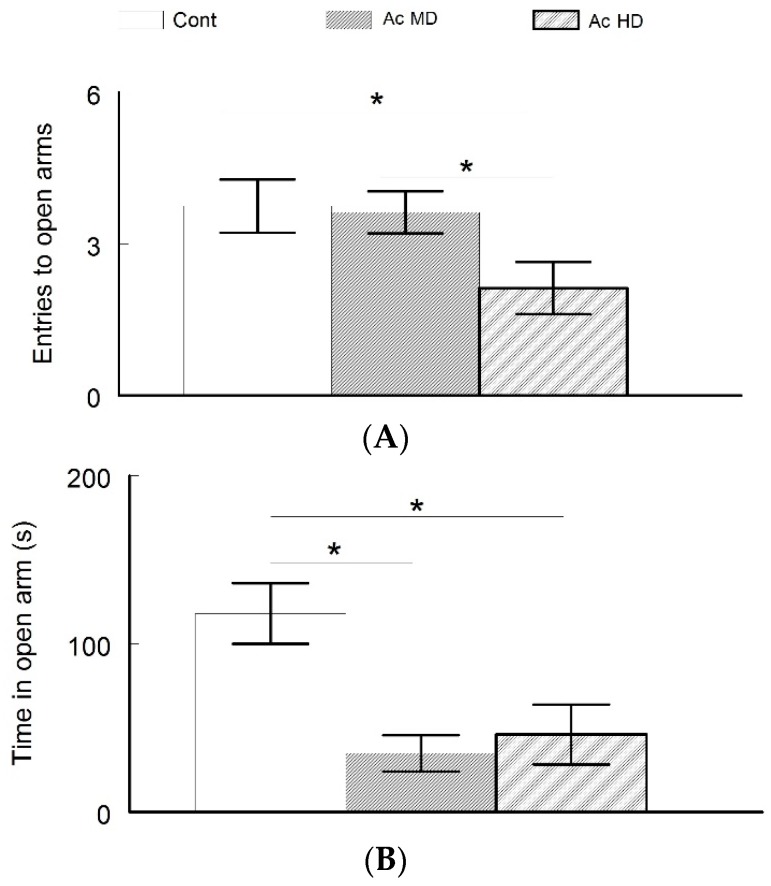
(**A**,**B**) Elevated plus maze test to assess anxiety. Although the Ac MD group (3.6 ± 0.42 s) entered the open arms more frequently than the Ac HD group (2.13 ± 0.52) (**A**), it spent the least amount of time in the open arms (34.9 ± 10.8) (**B**). Overall, the caffeine-treated mice were more anxious than the control group. Asterisks mark significant differences between the groups.

**Figure 4 behavsci-08-00065-f004:**
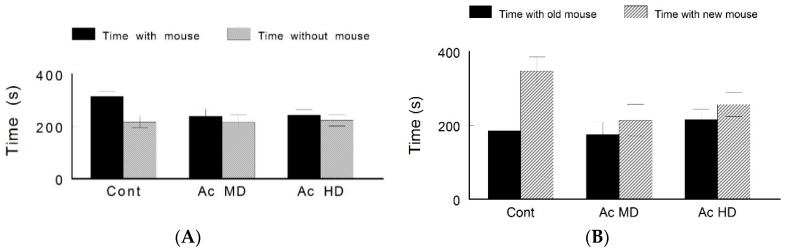
Three-chamber test to assess sociability and preference for social novelty. (**A**) Sociability (session 1: time spent in the chamber with the mouse versus the chamber without the mouse). (**B**) Preference for social novelty (session 2: time spent in the chamber with the old mouse versus the chamber with the novel mouse). Control animals showed normal sociability and preference for social novelty. Caffeine-treated mice showed decreased sociability and preference for social novelty.

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
