# Peer review of "Acute Administration of Caffeine: The Effect on Motor Coordination, Higher Brain Cognitive Functions, and the Social Behavior of BLC57 Mice"

_behavsci, 2018, doi:10.3390/bs8080065_

Round 1

Reviewer 1 Report

Summary: The manuscript by Almosawi et al. focuses on examining the effects of low-moderate or high doses caffeine intake on various behaviors. The authors found that while the low-moderate dose of caffeine appeared to improve motor and spatial memory, high doses of caffeine may have impaired these behaviors. However, exposure to both doses of caffeine increased anxiety-like behaviors and impaired social behavior. While these interactions are interesting, the authors do state that these types of studies have already been done, which reduces the enthusiasm for the study. Additionally, there are several data analyses and statistical issues that must be addressed in order to clearly interpret the data. Below are a comments/questions that if addressed would increase the impact of the paper:

How old were the animals when they were given caffeine and how long after the caffeine treatment were behavioral tests conducted?

Were different mice used for each behavioral test or were the same 8 mice used for all of the behavioral tests? Regardless, this information needs to be provided and the order of behavioral tests needs to be clearly stated.

The title for the y-axis on Figure 1 should be changed to “Latency to fall (s)”

The Morris Water Maze is primarily a test of spatial memory, not necessarily cognitive function. This should be reworded throughout the document.

In the results sections for the Morris Water Maze, it is not clear how the average values provided were calculated. Are they averages of the last 3 trials? This is particularly important because in the graph showing distance traveled there does not appear to be a difference between any of the groups.

Since subjects were repeated tested on the Morris Water Maze, then a repeated-measures ANOVA should have been used to compared performance across days.

There are several instances where the units are not provided or the incorrect unit is given. 1) Line 152, the unit for distance isn’t seconds as indicated in the main text; 2) line 154, the unit is missing for velocity; 3) line 155, you use "seconds" as your unit, but the graph states that it is % of time. The unit used needs to be consistent.

For Figure 2, A legend is necessary to describe what the symbols represent, or this information can be included in the figure legend.

For the elevated plus maze data, If the duration of the test was 10 minutes (600 seconds), how did the control animals spend more time in the open arms than the closed arms when the graph indicates that they only spent ~120 seconds in the open arms?

Also for the elevated plus maze figure, if there were significant difference between the caffeine groups and controls, symbols of significance need to be added.

The figure legend for figure 3 appears to be missing some words. Does it mean to say "it spent the least amount of time in the open arms"?

In figure 4, are the 2 bars in the control group significantly different from each other for both sessions 1 and 2?

Related to the question/comment in the methods section regarding whether the same animals were tested on all behavioral assays; if so, a discussion on the limitations of that needs to be included, such as testing interactions confounding or changing results on the subsequent tests.

How do the doses used in this study compare to doses used in previous studies? This could be a reason for the discrepant results.

In line 210, instead of saying "as they notably had", change to "although they had".

Line 219, add “in the” between “activity” and “striatum” to read “…activity in the striatum…”.

Line 221, change “stimulant” to “stimulating”.

Line 254, change “researches” to “research”.

Author Response

We appreciate very much the efforts of the reviewers, and we think all the points raised by them are valid and consequently, responding to them will make our article more solid and sound.

We will try to answer point by point all the matters raised, and indicate the changes that were made on the revised manuscript.

Reviewer # 2:

1-    How old were the animals when they were given caffeine and how long after the caffeine treatment were behavioral tests conducted?

Answer: This point was made more clear in the Methods section:

* Adult male BLC57 mice……………………. All tests were performed when the animals were six weeks of age.

2-    Were different mice used for each behavioral test or were the same 8 mice used for all of the behavioral tests? Regardless, this information needs to be provided and the order of behavioral tests needs to be clearly stated.

Answer: We thank the reviewer for reminding us to explain this important point. The following paragraph is added in the Materials and Methods section (animals):

Each of the groups was tested in Rota rod and water-maze in one day with three hours intervals. The elevated plus maze and three-chamber social apparatus tests were done the next day with same intervalsAll tests were performed when the animals were six weeks of age.

3-    The title for the y-axis on Figure 1 should be changed to “Latency to fall (s)”

 Answer: The figure was remade and requested title of Y-axis was changed.

4-    The Morris Water Maze is primarily a test of spatial memory, not necessarily cognitive function. This should be reworded throughout the document.

Answer: We agree that water maze is primarily a test for spatial memory, although the first 5 training sessions include also learning part. However, we replaced the term (brain cognitive function) into (spatial memory) to be more consistent as the reviewer suggested.

5-    In the results sections for the Morris Water Maze, it is not clear how the average values provided were calculated. Are they averages of the last 3 trials? This is particularly important because in the graph showing distance traveled there does not appear to be a difference between any of the groups.

Answer: The average of each point in the graph is an overall average of the performance of each animal (each given 5 trials), and then for all the animals. The data of *distance* may be slightly different than the *latency* when the animals may stop swimming then continue, or may swim with slightly different velocity. This happened very occasionally but affected little the *distance* results. That’s why in all such research, the *latency* is more specific.

6-    Since subjects were repeated tested on the Morris Water Maze, then a repeated-measures ANOVA should have been used to compared performance across days.

Answer: The ANOVA test was used.

7-    There are several instances where the units are not provided or the incorrect unit is given. 1) Line 152, the unit for distance isn’t seconds as indicated in the main text; 2) line 154, the unit is missing for velocity; 3) line 155, you use "seconds" as your unit, but the graph states that it is % of time. The unit used needs to be consistent.

 Answer: This is very correct, and we are sorry for the typo errors which were corrected in the new version.

8-    For Figure 2, A legend is necessary to describe what the symbols represent, or this information can be included in the figure legend.

 Answer: This is very valid point. The figure was redone to include the symbols.

9-    For the elevated plus maze data, If the duration of the test was 10 minutes (600 seconds), how did the control animals spend more time in the open arms than the closed arms when the graph indicates that they only spent ~120 seconds in the open arms?

 Answer: Thank you for alerting us to clarify this point. Actually when we said ( the control animals spend more time in the open arm than the closed arms), we did not mean (spend more time in the open than in the closed arms). We meant (spend more time in the opened arms compared to the other groups). This was also corrected in the Results by the following senrtence:

* Control animals showed less anxiety- like behavior by spending more time in the open arms compared to the other groups during the test.

10- Also for the elevated plus maze figure, if there were significant difference between the caffeine groups and controls, symbols of significance need to be added.

Answer: The figure was re done and the significance symbols were added.

10- The figure legend for figure 3 appears to be missing some words. Does it mean to say "it spent the least amount of time in the open arms"?

Answer: This is true and the sentence was corrected accordingly.

* Although Ac MD group (3.6 ± 0.42 s) entered the open arm more frequent than the Ac HD group (2.13 ± 0.52) (Fig 3A), it spent the least amount of time in the open arms (34.9 ± 10.8) (Fig 3B).*

11- In figure 4, are the 2 bars in the control group significantly different from each other for both sessions 1 and 2?

Answer: Fig 4A represents the time spent by the tested animal in the chamber which contained another mouse. So the tested mouse had to choose to stay in a chamber with an animal  or in a chamber without. The setup of session 2 of the experiment (Fig 4B), the tested animal had to choose between staying in the chamber with the old animal (known before) or rather in the chamber which contained a new animal.

12- Related to the question/comment in the methods section regarding whether the same animals were tested on all behavioral assays; if so, a discussion on the limitations of that needs to be included, such as testing interactions confounding or changing results on the subsequent tests.

Answer: The sequence by which we tested the animals was added which ensures that there was minimal effect of one test on the other. The sequence was as follow:

Each of the groups was tested in Rota rod and water-maze in one day with three hours intervals. The elevated plus maze and three-chamber social apparatus tests were done the next day with same intervalsAll tests were performed when the animals were six weeks of age.

13- How do the doses used in this study compare to doses used in previous studies? This could be a reason for the discrepant results.

Answer: The doses in ml, and also in mg/kg are now mentioned (in the section of Materials and methods/ Animals, and they are comparable to other research work. A reference was added as well.

* The animals were divided into the following: control group (n=8, were given normal water), acute moderate dose (n=8, were given moderate dose caffeine (0.1g/L i.e 20 mg/kg) for 1 week, Ac MD), acute high dose (n=8, were given high dose caffeine for 1 week, Ac HD 1g/L i.e 200 mg/kg). These doses are comparable to other previous studies[25].*

14- In line 210, instead of saying "as they notably had", change to "although they had".

 Answer: Corrected as requested.

14- Line 219, add “in the” between “activity” and “striatum” to read “…activity in the striatum…”.

 Answer: Done

15- Line 221, change “stimulant” to “stimulating”.

 Answer: Done

16- Line 254, change “researches” to “research”.

Answer: Done.

We finally hoping that our edited manuscript is now suitable for publication.

Reviewer 2 Report

This is, in summary, an interesting manuscript aimed to investigate the effects of excessive caffeine consumption on health in a preclinical model. The authors found that mice taking moderate caffeine doses could stay more time on the rotating rod before they fall when compared with the control group and the high dose group. Water maze test for learning and memory showed better performance of mice receiving moderate caffeine dose when compared to the other groups. In addition, animals that were administered moderate as well as high doses of caffeine showed no sociability and no preference for social novelty in the three-chamber test used to test the social behavior. Moreover, in elevated plus maze, control animals demonstrated no anxiety-like behavior while mice administered with caffeine were both showing anxiety-like behaviors.

The authors may find as follows my main comments/suggestions.

First, when in the Introduction section, the authors correctly stated that nsomnia or sleep deprivation results in the lower retention rate of memories due to the neural connections, they could also mention the link between insomnia and negative outcome such as suicidal behavior. Specifically, research demonstrated that patients with sleep alterations, particularly insomnia, are at an increased risk of experiencing suicidal ideation and/or making a suicide attempt. Although the association between insomnia and suicidal behavior is not the main topic of the present paper, the authors might refer, at least briefly, to the mentioned topic citing the paper which has been published on International Journal of Clinical Practice (67(12):1311-1316) in 2013.

In addition, within the same section the assumption that the higher caffeine dose may have depressive effects needs to be specified in a more detailed manner and supported by adequate references. Conversely, the authors reported only one reference supporting this notion which is mainly focused on social anxiety disorder.

Moreover, given that the main aims of this paper have been extensively proposed by the authors, the specific hypotheses underlying the study objectives should be adequately reported as well.

Furthermore, when the authors referred, within the Discussion section, to “benzodiazepine receptors”, i believe that they should mention more appropriately the gabargic receptors, the subunits of which represent the real targets of these componds. Here, changes to the main text are requested. Similarly, when they reported that in relation to higher brain functions and social behavior, antagonist of adenosine receptors is the most important mechanism, one or more references are needed.

Another additional aspect needs further details/information by the authors. As they reported that higher caffeine doses are associated with a deterioration in motor/cognitive functions as a result of an increasingly progressive anxious behavior, the mechanisms underlying this should be, at least partially, explained within the main text. Similarly, when the authors stated that higher caffeine doses are linked to anxiogenic effects. Here, more information are needed.

Importantly, the most relevant shortcomings/limitations of this study need to be cited and discussed as their description is completely lacking according to the current version of the paper.

Author Response

We appreciate very much the efforts of the reviewers, and we think all the points raised by them are valid and consequently, responding to them will make our article more solid and sound.

We will try to answer point by point all the matters raised, and indicate the changes that were made on the revised manuscript.

Reviewer #1:

1- when in the Introduction section, the authors correctly stated that insomnia or sleep deprivation results in the lower retention rate of memories due to the neural connections, they could also mention the link between insomnia and negative outcome such as suicidal behavior…..ect

Answer: We think this reference is important and it is added in the first paragraph in the Introduction section.

* Sleep deprivation, in general, is associated with deterioration of memory [6] as well as other negative outcome such as suicidal behavior [7]*

2- In addition, within the same section the assumption that the higher caffeine dose may have depressive effects needs to be specified in a more detailed manner and supported by adequate references. Conversely, the authors reported only one reference supporting this notion which is mainly focused on social anxiety disorder.

Answer: The following sentence was added with three new relevant reference;

* [12] A significant correlation was found between the caffeine dose and the level of depressive illness [13]. Data from other researches shows a direct correlation between the adenosine homeostasis and mood disorders. Patients with major depression were found to have lower levels of serum adenosine deaminase compared to the control group, with a negative correlation between the enzyme activity and the severity of depression.[14]

3- Moreover, given that the main aims of this paper have been extensively proposed by the authors, the specific hypotheses underlying the study objectives should be adequately reported as well.

Answer: We think this is a very important point and we thank the reviewer for mentioning. We added the following paragraph at the end of Introduction section:

* This study aimed to measure the effect of caffeine on learning, memory, anxiety and the social behavior on mice. We hypothesize that administering moderate doses of caffeine increases the higher brain functions and locomotion of mice. Memory enhancement could be due to inhibition of adenosine A1 receptors that strongly inhibit the release of acetylcholine from pyramidal hippocampal neurons, while locomotor enhancement could be attributed to the multiple effects of caffeine on skeletal muscle contraction (by either modulating the calcium homeostasis in the muscle fibers and/or increasing the sensitivity of myofilaments to calcium ions). Furthermore, caffeine treated mice will display anxiogenic behavior and a decrease in social parameters, which can be caused by blockade of benzodiazepine binding sites on GABAA receptors, stimulation of central noradrenergic activity or antagonism of adenosine receptors.

.*

4- Furthermore, when the authors referred, within the Discussion section, to “benzodiazepine receptors”, i believe that they should mention more appropriately the gabargic receptors, the subunits of which represent the real targets of these componds. Here, changes to the main text are requested. Similarly, when they reported that in relation to higher brain functions and social behavior, antagonist of adenosine receptors is the most important mechanism, one or more references are needed.

Answer: Part of this point is answered in the previous paragraph (hypothesis).

Regarding the mechanism by which adenosine receptors antagonism affecting higher brain function and social behavior the following paragraph with references was added in the first paragraph of the second page of Discussion*.

* Adenosine receptors knockout studies demonstrated defects in cognition, memory and social behavior [43]. In addition, synaptic plasticity of the hippocampus as long-term potentiation (LTP) and depression (LTD) and higher brain functions were shown to be modulated by adenosine receptors and consequently by adenosine receptors antagonism [44].

5- Another additional aspect needs further details/information by the authors. As they reported that higher caffeine doses are associated with a deterioration in motor/cognitive functions as a result of an increasingly progressive anxious behavior, the mechanisms underlying this should be, at least partially, explained within the main text. Similarly, when the authors stated that higher caffeine doses are linked to anxiogenic effects. Here, more information are needed. Importantly, the most relevant shortcomings/limitations of this study need to be cited and discussed as their description is completely lacking according to the current version of the paper.

Answer: This is very important point and we thank the reviewer for raising it. The following paragraph is added (third paragraph in Introduction) with the relevant references.

* Due to the effect of caffeine on Adenosine receptors and on multiple neurotransmitters that lead to arousal, high doses of caffeine can cause dysfunctional arousal state that causes sleep disturbances that include alteration of sleep quality, latency and total sleep time. [15]

Although that the certain mechanism is not yet fully understood, multiple researches successfully linked sleep deprivation with anxiety disorder.[16][17]

Round 2

Reviewer 1 Report

The authors have addressed all of my original concerns. However, the revision made in response to my first question on the 1st review is not correct. The authors state that the animals used in the study were 6 weeks old and that this corresponds to using adult animals. However, there is a large literature indicating that 6 weeks or ~postnatal day 42 is still adolescence in rodents. Adulthood typically begins ~postnatal day 60-70. This is an important consideration that should be corrected in the methods, which would also be worth discussing in the discussion section regarding the potential implications of the current results in adolescence versus what other studies may have shown using caffeine exposure in true adults.

Author Response

We thank very much the continuous support and positive remarks and suggestion.

Concerning the age of the animals... we corrected the manuscript in the following way:

We substituted the word (adult) in the (Materials and methods/ section Animals) into (Adolescent).

We added the following paragraph in the (Discussion) section, at the end of the second paragraph of the least page in Discussion (Shaded in red):

However, it should be mentioned that our results apply for caffeine use on adolescent male mice. Different outcome results cannot be excluded by studying the effects of caffeine on female mice or younger or older age groups.

We hope that the manuscript is suitable now in its revised version for publication.

Reviewer 2 Report

In the revised manuscript,the authors addressed most of the major questions raised by the Reviewers improving the main structure of the paper. I have no further additional comments.

Author Response

We thank very much the valuable remarks and suggestions to make our manuscript in most acceptable form.